# Assessment of Raman Spectroscopy for Reducing Unnecessary Biopsies for Melanoma Screening

**DOI:** 10.3390/molecules25122852

**Published:** 2020-06-20

**Authors:** Yao Zhang, Austin J. Moy, Xu Feng, Hieu T. M. Nguyen, Katherine R. Sebastian, Jason S. Reichenberg, Claus O. Wilke, Mia K. Markey, James W. Tunnell

**Affiliations:** 1Department of Biomedical Engineering, The University of Texas at Austin, Austin, TX 78712, USA; zhangyao1994@utexas.com (Y.Z.); austin.moy@gmail.com (A.J.M.); xu.feng@utexas.edu (X.F.); hieung@utexas.edu (H.T.M.N.); mia.markey@utexas.edu (M.K.M.); 2Department of Medicine, The University of Texas at Austin, Austin, TX 78712, USA; KrSebastian@ascension.org (K.R.S.); JReichenberg@ascension.org (J.S.R.); 3Department of Integrative Biology, The University of Texas at Austin, Austin, TX 78712, USA; wilke@austin.utexas.edu; 4Department of Imaging Physics, The University of Texas MD Anderson Cancer Center, Houston, TX 77230, USA

**Keywords:** melanoma, skin screening, Raman spectroscopy, classification, specificity

## Abstract

A key challenge in melanoma diagnosis is the large number of unnecessary biopsies on benign nevi, which requires significant amounts of time and money. To reduce unnecessary biopsies while still accurately detecting melanoma lesions, we propose using Raman spectroscopy as a non-invasive, fast, and inexpensive method for generating a “second opinion” for lesions being considered for biopsy. We collected in vivo Raman spectral data in the clinical skin screening setting from 52 patients, including 53 pigmented lesions and 7 melanomas. All lesions underwent biopsies based on clinical evaluation. Principal component analysis and logistic regression models with leave one lesion out cross validation were applied to classify melanoma and pigmented lesions for biopsy recommendations. Our model achieved an area under the receiver operating characteristic (ROC) curve (AUROC) of 0.903 and a specificity of 58.5% at perfect sensitivity. The number needed to treat for melanoma could have been decreased from 8.6 (60/7) to 4.1 (29/7). This study in a clinical skin screening setting shows the potential of Raman spectroscopy for reducing unnecessary skin biopsies with in vivo Raman data and is a significant step toward the application of Raman spectroscopy for melanoma screening in the clinic.

## 1. Introduction

Skin cancer is the most commonly diagnosed cancer in the United States [1]. For melanoma, which accounts for the vast majority of skin cancer deaths, there are an estimated 100,350 new cases and 6850 deaths in the US for 2020 [1]. Early detection is a key strategy to saving these lives.

The current gold standard for melanoma detection is as follows [2]. During skin screening, experienced dermatologists assess a concerning skin lesion visually and determine whether to conduct a skin biopsy, which involves surgical excision of the skin lesion. Then, the excised lesion is sent to a histopathology lab for analysis, and it may take up to two weeks to get the final diagnosis result for the biopsied lesion [3]. This process is invasive, time-consuming, and costly.

Most biopsied lesions turn out to be benign pigmented lesions that did not need to be removed. The number needed to treat (NNT), or the number needed to biopsy (NNB), a metric for evaluating the number of unnecessary biopsies, is calculated by dividing the total number of biopsied lesions by the number of histologically proven melanomas. Prior studies report a NNT of 6.0–30 for melanoma in different skin screening settings [4,5,6,7,8,9], which means 83.3–96.7% of biopsies are unnecessary. These unnecessary skin biopsies create significant time burden and financial burden for both patients and the healthcare system. Therefore, a method is urgently needed to reduce the number of unnecessary biopsies while not missing any additional melanomas. Especially, novel tools are needed to help dermatologists distinguish pigmented lesions from melanoma because melanomas are uncommonly diagnosed from nonpigmented lesions [5].

Raman spectroscopy (Raman) is a non-invasive, fast, and inexpensive technology that has promise for skin cancer detection [4,10,11,12,13,14,15,16,17,18,19,20,21,22,23,24,25,26,27,28]. Three prior studies are particularly relevant to the current investigation. First, Lui et al. reported that they could separate malignant melanoma (*n* = 44) from biopsied nonmelanoma pigmented skin lesions (*n* = 81) with an area under the receiver operating characteristic curve (AUROC) of 0.833 (95% CI, 0.761–0.906) based on leave one out cross validation of in vivo Raman spectra collected in a high risk screening clinic [22]. Similarly, our group’s prior trial conducted in a high risk screening clinic showed that in vivo Raman spectra enabled classification of melanoma versus pigmented lesions with sensitivity and specificity of both 100% for a data set of 12 melanoma versus 17 pigmented lesions [23,25,26,27]. Thus, both Lui et al. and our prior trial demonstrated that in vivo Raman could reduce unnecessary biopsies for melanoma diagnosis in high risk screening clinics. However, a limitation of these prior studies is that the NNT in a typical skin screening setting (6.0–30 [4,5,6,7,8,9]) is usually much larger than the NNT in a high risk screening clinic (2.4–2.8 [22,23]). Third, Santos et al. conducted an ex vivo study of 174 freshly excised melanocytic lesions in a typical skin screening setting and showed that Raman could have improved clinical diagnosis of early-stage cutaneous melanoma from NNT of 6.0–2.7 while maintaining 100% sensitivity [4]. A constraint of the work of Santos et al. is that it relied on Raman of ex vivo samples rather than in vivo samples.

To assess the potential of Raman to decrease the NNT for in vivo melanoma diagnosis in a skin screening setting, we collected Raman spectral data from 52 patients with lesions that the physician was concerned may be melanoma [2]. During clinical skin screening, we acquired measurements on all lesions that were about to be biopsied. Conducting this work in vivo in a skin screening setting is important because we aim to use Raman spectroscopy and our data analysis models to provide a “second opinion” for lesions that dermatologists are considering for biopsy during skin screening. Our goal is to reduce the number of unnecessary biopsies (improve specificity and true negative rate) while identifying melanoma lesions accurately (high sensitivity, true positive rate) during melanoma screening.

## 2. Results

Figure 1a shows that the AUROC for classifying melanoma vs. pigmented lesions is 0.903. If our recommendations based on Raman spectroscopy had been enacted, approximately 58.5% of the biopsies on pigmented lesions could have been avoided while accurately detecting all melanoma lesions in the data set (sensitivity of 100%). The blue shaded area shows the 95% confidence interval for the ROC, which is wide because there are only seven melanoma lesions in our dataset.

To address the concern that the promising ROC curve could be due to chance given the limited sample size, randomization tests were conducted by assigning seven lesions randomly out of 60 total lesions to the “melanoma” category. Figure 1b shows that the random chance for the AUROC to be greater than 0.9 is very small (1%), which suggests that the observed high AUROC for our data set is mostly likely due to real differences between melanoma and pigmented lesions.

Principal Component (PC) 2, 7, 8, 9, 10, and 11 were used to create the logistic regression classifier to generate the AUROC of 0.903. The first PC, which captures the most variance of original dataset, was not valuable for our classification task. The combination of PCs that generated the highest AUROC is reported here to demonstrate the promise of using Raman data for this classification task of melanoma versus pigmented lesions. Likewise, for the randomization tests, the highest AUROC was reported among all possible combination of PCs to ensure fair comparison.

Table 1 shows the prediction accuracy summary. In principle, our model could have eliminated the need for 58.5% of biopsies while still detecting all of the melanomas that the dermatologists detected. The NNT for melanoma could have been decreased from 8.6 (60/7) to 4.1 (29/7) if dermatologists followed the “second opinion” biopsy recommendation from Raman spectroscopy based model. Our results demonstrate the promise of using in vivo Raman for classifying melanoma and pigmented lesions to reduce unnecessary biopsies in a typical skin cancer screening setting.

## 3. Discussion

Understanding the biological alterations that lead to different Raman spectra is important, and we have recently published a detailed report of the underlying biophysical basis of Raman spectra in skin tumors [26]. We found that at least eight primary tissue constituents can independently contribute to the measured Raman spectrum and include a combination of lipids (triolein and ceramide), proteins (collagen, keratin, elastin, and melanin), nucleic acids (DNA), and water [25]. The primary discriminating biophysical differences between pigmented lesions and melanoma result from proteins (collagen) and lipids (triolein). While it is difficult to interpret the contributions of these various constituents in a single spectrum, one can appreciate the reduction in the CH2/CH3 bands associated with lipids and the shift in the Amide III bands associated with proteins, as shown in Appendix A. Due to the complexity in determining the biophysical contributions to the overall spectrum, statistical methods are widely used for spectral diagnosis and we have chosen to use PCA in this case.

In our study, PCA was used because it is adept at the dimensional reduction of spectral data and can be used to calculate uncorrelated PCs to represent the original spectral data. A limited number of highest varying PCs could be used to create classifiers. Alternative methods could be potentially applicable for classification using high dimensional data, such as the network-based regularization method for the logistic model [29]. We did not report the one-step regularized logistic regression method because different large ranges of wavenumbers were helpful for our classification problem and it was challenging to select a limited number of wavenumbers to overcome the overfitting concern while achieving adequate classification results.

Even if multiple measurements were taken on each lesion, those measurements should be considered independently because of the biological variation in each lesion. Measuring a single point on a lesion runs a high risk of missing cancer. Therefore, we used all measurements of each lesion when applying PCA on the dataset. In other words, one-point measurement on a cancerous lesion might turn out to be normal and multiple measurements are helpful to capture all the cancers. The number of measurements should depend on the size of the lesion since the measurements should cover the majority of the lesion area. It is possible that lesions with more measurements might contribute more than the lesions with less measurements to our PCA results, but this is not a concerning issue because our cross-validation classification results provided high accuracy, which is significantly higher than randomization tests with the same model training process.

Our study was limited by the small size of our dataset. It is unlikely that a sample of 60 lesions, including only seven melanoma lesions, fully encompassed the true biological diversity of human skin. Moreover, due to the small sample size, “leave-one-lesion-out” cross-validation was used for model development and assessment, which is less conclusive than a design employing independent training, validation, and testing sets. Additionally, we reported the single combination of PCs that yielded the numerically highest AUROC, but several alternate models would provide statistically indistinguishable classification performance given the sample size. However, additional analyses were undertaken to mitigate these limitations arising from the sample size. Especially, we conducted randomization tests in which the same analysis pipeline was performed but with seven lesions randomly assigned to the “melanoma” category. This randomization analysis achieved mostly low AUROCs, demonstrating that the observed high AUROC for our data set is most likely due to real differences between melanoma and pigmented lesions. Of course, a future study with large independent training, validation, and testing datasets of in vivo Raman obtained in a typical skin cancer screening setting would be important to further validate the use of Raman for reducing unnecessary biopsies for melanoma diagnosis.

Although our results demonstrate the feasibility of reducing unnecessary biopsies using our in vivo Raman-based tool in a typical skin cancer screening setting, we have not asked dermatologists to hypothesize about whether they would have changed their decision to biopsy if our results suggested that a biopsy was not needed. Research in other areas of medical imaging demonstrates that computer aided detection or diagnosis recommendations do not necessarily change a physician’s actions [30]. Therefore, future studies are needed on the acceptability of our tool in dermatological practice.

## 4. Materials and Methods

### 4.1. Optical Instrument System and Dataset

The clinical optical spectroscopy system has been previously described [2,31,32]. Briefly, light is emitted to and collected from the skin via a custom-designed handheld fiber probe that performs Raman measurements. The light source for Raman excitation is an 830 nm diode laser (Ondax, Monrovia, CA, USA). The collected light travels to a custom configured Raman spectrometer with a full width at half maximum (FWHM) of 19 cm^−1^ (Stroker model from Wasatch Photonics, Morrisville, NC, USA), and LabVIEW (NI, Austin, TX, USA, 2015) is used for all components control and data acquisition. Figure 2 shows the optical instrument system and the probe.

Clinical spectra were collected at the Seton Healthcare Family dermatology clinic in accordance with a human subjects research protocol approved by the Institutional Review Board at The University of Texas at Austin. All participants received informed consent prior to participation in this study. Before each data acquisition, the handheld fiber probe was disinfected and cleaned with sterile alcohol pads to ensure patient safety. Raman integration time was 2 seconds, and the detection depth of measurement was less than 1 mm and depends on the tissue optical properties.

Table 2 provides a summary of the clinical data used in this manuscript. Between 23 March 2016 and 29 July 2017, spectra were acquired from 52 volunteer participants on 60 lesions that a dermatologist was concerned may be melanoma. A minimum of two spectral measurements was collected on each lesion, and larger lesions had additional measurements depending on the size of the lesion to ensure adequate sampling of the lesion surface area. In total, 158 measurements were acquired on 53 pigmented lesions and 27 measurements were acquired on 7 melanoma lesions. Hence, there were 53 unneeded biopsies of pigmented lesions in this clinical dataset, and the NNT for melanoma was 8.6 (60/7). A more detailed clinical data summary is provided in Appendix A. Note that even pathologists debate the differences in different types of nevi and there are no accurate “gold standard” as ground truth other than pathology examination, so we focused on the difference between pigmented lesions and melanoma here. A summary of lesion information evaluated by Raman is also provided in Appendix A.

### 4.2. Data Analysis Pipeline

Raw Raman spectra pre-processing was described in previous studies [2,23,27]. Briefly, the raw Raman spectra underwent wavenumber calibration, dark noise background removal, cosmic ray removal, smoothing, and a fifth-order polynomial fitting [33] to remove tissue fluorescence background. Spectral intensity response was calibrated using a LS-1-CAL calibrated tungsten halogen lamp (Ocean Optics, Dunedin, FL, USA). The effective spectral range was 800–1790 cm^−1^. Data were normalized by scaling the area under the curve to equal one [22]. Appendix A shows representative Raman spectra of melanoma and pigmented lesions.

Figure 3 shows the data analysis pipeline, which was conducted in MATLAB (R2019b, MathWorks, Natick, MA, USA). We used principal component analysis (PCA) and built logistic regression classifiers with leave one lesion out cross validation, and receiver operating characteristic (ROC) curves were used for accuracy evaluation. Normalized Raman data were used as inputs for PCA. PCA was used because there were 1980 wavenumbers between 800 and 1790 cm^−1^ for our Raman data and adjacent wavenumbers were correlated. PCA is able to reduce the dimensionality dramatically by providing the principal components (PCs), which capture the most variance of the original data. Logistic regression models are appropriate for binary classification tasks. Leave one lesion out cross validation was used to validate the performance of the PCA and logistic regression models by leaving one lesion out each time to train the models. PCA was applied to the training dataset, and the same transformation using the calculated coefficients (PCA rotation object) from the training dataset was applied to the test dataset. A subset of PCs was used as input features for logistic regression models and the probability for each left out biopsy was calculated in the whole cross validation process. ROC curves were then calculated based on the posterior probabilities of all lesions and used to evaluate the performance of classification. The ROC curve with confidence intervals was plotted in R (Version 1.2.5033, RStudio, Boston, MA, USA).

To produce the subsets of PCs compared in this study, all possible combinations of PCs were generated from the first 15 PCs while limiting the number of PCs to 6. The scree plot was referred to in order to determine the number of potentially selected PCs. The most significant 15 PCs accounted for the majority (90.22%) of the variance in original data while the PC 15 explains 0.4361% of the variance. The “one in ten” rule informed our choice of the number of features to consider [34]. Briefly, in order to have at least 10 data points for each model parameter, we limited the number of PCs in the model to 10% of the total number of lesions (60), i.e., 6 PCs. The subset of PCs that yielded the highest AUROC was reported here. We acknowledge that several subsets of PCs yielded similar AUROCs that are not statistically significantly different from the highest AUROC.

The ROC curves were calculated by treating each lesion as an experimental observation, described as per lesion analysis in Lim et al. [23]: only if all spectra measurements from one site are classified as normal is this site classified as normal; otherwise, this site is classified as cancerous if at least one spectral measurement is classified as cancerous. The higher area under the ROC curve is, the more predictable the model is. An AUROC of 1 means perfect classification.

Our classification task is melanoma versus pigmented lesions because melanoma should be biopsied while pigmented lesions should not be biopsied.

## 5. Conclusions

To conclude, in vivo Raman spectra collected in a typical skin cancer screening setting were used for classifying melanoma versus pigmented lesions. Principal component analysis and logistic regression models with leave one lesion out cross-validation showed the promise of using Raman measurements for skin cancer screening to reduce unnecessary biopsies on pigmented lesions while correctly identifying all melanoma lesions. Our model achieved an area under the ROC curve (AUROC) of 0.903, and 58.5% of biopsies of pigmented lesions could have been avoided while still accurately recommending biopsy of all the melanomas identified by the dermatologist. The number needed to treat (NNT) for melanoma could be decreased from 8.6 to 4.1 if dermatologists followed the “second opinion” from our Raman-based during skin screening. Our work serves an important step of promoting the application of Raman for melanoma screening in the clinic.

## Figures and Tables

**Figure 1 molecules-25-02852-f001:**
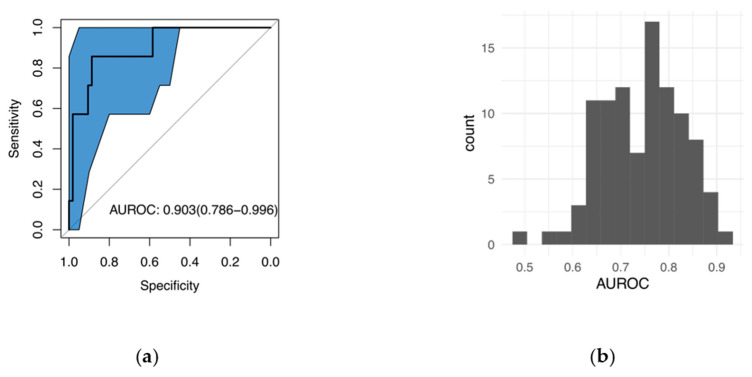
(**a**) Receiver operating characteristic (ROC) curve for classifying melanoma vs. pigmented lesions. The area under the ROC curve (AUROC) of 0.903 means high accuracy in distinguishing melanoma from pigmented lesions. The blue shade shows the 95% confidence interval for the ROC. (**b**) The histogram of AUROC from 99 randomization tests, where 7 out of 60 lesions were randomly assigned to the “melanoma” group. There is only a small chance (1%) that the observed AUROC is greater than 0.9 for the randomization tests, which suggests that the high AUROC is mostly likely due to real differences between melanoma and pigmented lesions.

**Figure 2 molecules-25-02852-f002:**
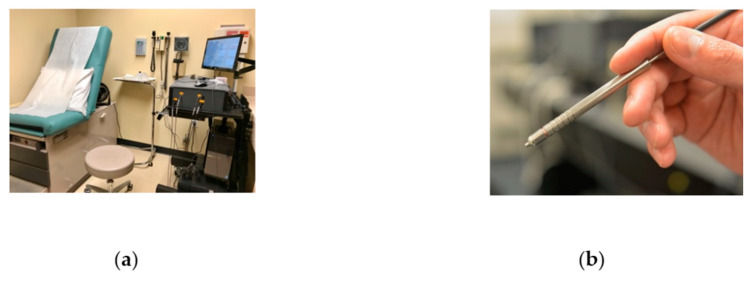
(**a**) Optical instrument system for clinical data acquisition in the clinic examination room. (**b**) The handheld fiber probe enables acquisition of Raman spectral data.

**Figure 3 molecules-25-02852-f003:**
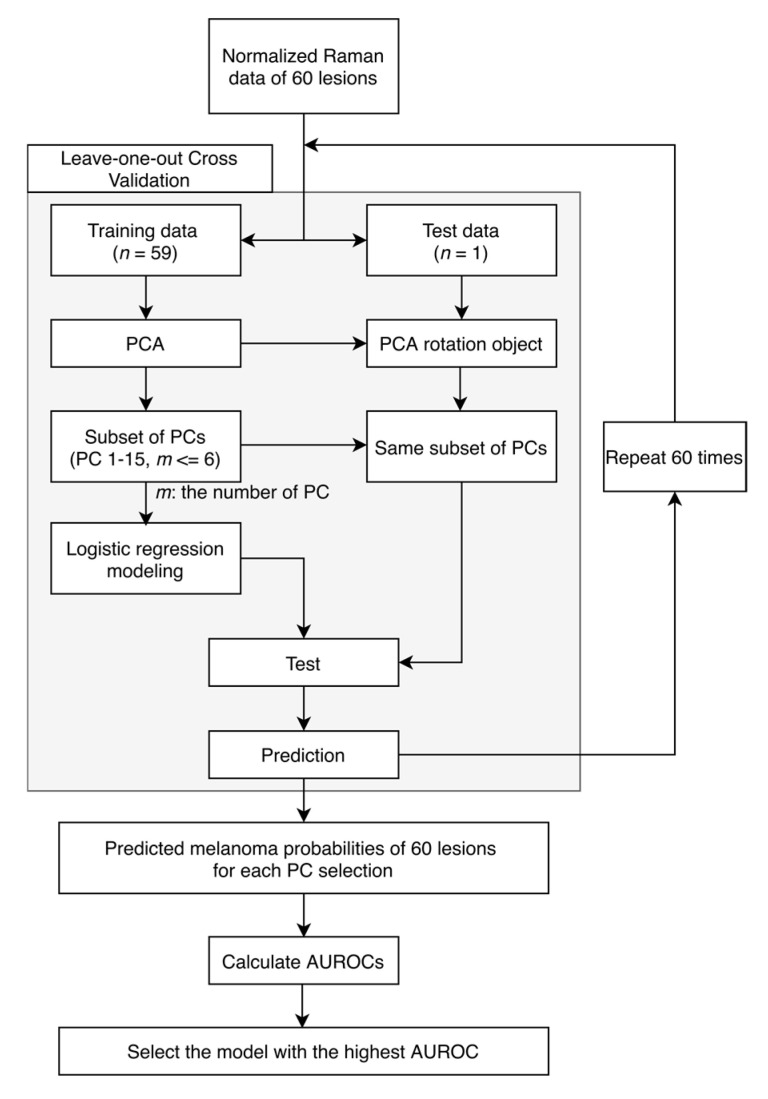
Data analysis pipeline. Principal component analysis (PCA) and logistic regression model with leave one lesion out cross validation are applied to classify melanoma and pigmented lesions based on normalized Raman spectral data. AUROC shows the classification accuracy.

**Table 1 molecules-25-02852-t001:** Prediction accuracy summary.

Lesion Type	Lesions	Correct Predictions (%)	False Predictions	Potential Biopsies
Pigmented Lesions	53	31 (58.5%)	22	22
Melanoma	7	7 (100%)	0	7
Total	60	38	22	29

**Table 2 molecules-25-02852-t002:** Clinical data summary.

Lesion Type	Patients	Lesions	Measurements
Pigmented lesions	51	53	158
Melanoma	6	7	27

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
