# Peer review of "Assessment of Raman Spectroscopy for Reducing Unnecessary Biopsies for Melanoma Screening"

_molecules, 2020, doi:10.3390/molecules25122852_

Round 1

Reviewer 1 Report

Zhang et al. has conducted classification analysis using Raman spectral data to demonstrate the importance of Raman spectroscopy for clinical screening of melanoma. Here are my comments.

It has been mentioned that the sample size of lesions is very limited. It looks like a “large dimensionality, small sample size ” problem, but the number of biological features (the dimension) is not provided—only sample size (n=60) is known. Without such information, it is not clear why PCs are created as features/predictors in the logistic regression models. A more natural approach is penalized classification. For example, the network-based regularization method for logistic model (such as Ren et al., PMID: 28511641 along with the R package regnet) can be readily adopted to achieve the classification task in one step, instead of the two step approach (PCA+ logistic regression). Such a one step method yields excellent identification results in terms of AUROC, as shown in Ren et al. As for PCA, it is an unsupervised method without using the response (the class label). So why extracting PCA is of interest as the task is binary classification? Wouldn’t it be more intuitive to use a supervised method directly? Please provide more justifications on the two step analysis and more discussions on alternative methods that are potentially applicable.

Line 197 –201, why a scree plot is not used to determine the number of selected PCs? The 10% cutoff seems arbitrary. Why only the first 15 PCs are chosen? Instead of using the leave one out cross validation, if the original data is randomly split as 3/4 for training and the rest 1/4 for testing, is the current model still the best one? If misclassification rate is adopted as the response for training the model, the leave one out might cause instability (see next comment). 

In addition sample size issues, any explanation for the unstable zig zag pattern of the ROC in Figure 1? Usually, ROCs have much more smooth curves. 

I am not against the randomization analysis but it will be more convincing if the classifier performs well on an independent testing data.

Line 135. “…testing sets datasets …”. “Sets” is redundant.

Is the figure on the first page misplaced?

Reviewer 2 Report

This short study involves Raman spectroscopy investigations targeted on melanoma diagnosis. It is not clear whether these findings could be accepted in  dermatological practice because Raman spectroscopy is so far not standard method. However, presented results could  give good foundation for further more detailed experiments.

What I completely miss is detailed discussion on the Raman spectra - why the spectra (fig S1) are different for the two selected samples. What are the skin changes which translate visible difference to the spectra for MM and PL? Why for instance is the Amide III peak different for the two samples??? Authors should consult dermatologist in order to answer these questions which in turn could elucidate the findings from the "biological" point of view.

Also I would like to see detailed explanation how authors normalized  the area under the curve using their data. Why this type of normalization was selected?  It is not explained here.

Fig S1 - Raman spectra - is still with very high noise which could potentially lead to large  errors in PCA calculations - could authors comment on this? 

Round 2

Reviewer 1 Report

I thank the authors for addressing all my comments.